# 3D Porous MXene (Ti_3_C_2_T_x_) Prepared by Alkaline-Induced Flocculation for Supercapacitor Electrodes

**DOI:** 10.3390/ma15030925

**Published:** 2022-01-25

**Authors:** Weihua Chen, Jiancheng Tang, Peidong Cheng, Yunlong Ai, Yi Xu, Nan Ye

**Affiliations:** 1School of Material Science and Engineering, Nanchang University, No. 999, Xuefu Avenue, Nanchang 330031, China; cweihua@126.com (W.C.); nuaaxuyi@163.com (Y.X.); yenan870831@163.com (N.Y.); 2School of Materials Science and Engineering, Nanchang Hangkong University, No. 696, South Fenhe Avenue, Nanchang 330063, China; cpdong@126.com (P.C.); ayunlong@126.com (Y.A.)

**Keywords:** Ti_3_C_2_T_x_, alkaline treatment, annealing treatment, 3D porous structure, supercapacitors

## Abstract

2D layered MXene (Ti_3_C_2_T_x_) with high conductivity and pseudo-capacitance properties presents great application potential with regard to electrode materials for supercapacitors. However, the self-restacking and agglomeration phenomenon between Ti_3_C_2_T_x_ layers retards ion transfer and limits electrochemical performance improvement. In this study, a 3D porous structure of Ti_3_C_2_T_x_ was obtained by adding alkali to a Ti_3_C_2_T_x_ colloid, which was followed by flocculation. Alkaline-induced flocculation is simple and effective, can be completed within minutes, and provides 3D porous networks. As 3D porous network structures present larger surface areas and more active sites, ions transfer accelerates, which is crucial with regard to the improvement of the superior capacitance and rate performance of electrodes. The sample processed with KOH (K-a-Ti_3_C_2_T_x_) exhibited a high capacity of approximately 300.2 F g^−1^ at the current density of 1 A g^−1^. The capacitance of the samples treated with NaOH and LiOH is low. In addition, annealing is essential to further improve the capacitive performance of Ti_3_C_2_T_x_. After annealing at 400 °C for 2 h in a vacuum tube furnace, the sample treated with KOH (K-A-Ti_3_C_2_T_x_) exhibited an excellent specific capacitance of approximately 400.7 F g^−1^ at a current density of 1 A g^−1^, which is considerably higher than that of pristine Ti_3_C_2_T_x_ (228.2 F g^−1^). Furthermore, after 5000 charge–discharge cycles, the capacitance retention rate reached 89%. This result can be attributed to annealing, which can further remove unfavourable surface groups, such as –F or –Cl, and then improve conductivity capacitance and rate performance. This study can provide an effective approach to the preparation of high-performance supercapacitor electrode materials.

## 1. Introduction

With the increase in environmental problems, the rapid development of new energy equipment, such as electric vehicles, has led to an urgent requirement for energy-storage devices with high-energy density and high cycle stability [1,2]. The key to solving these problems is to construct efficient and safe energy-storage devices to store renewable energy and to obtain productivity effectively [3]. Supercapacitors are electrochemical capacitors, which have a higher capacitance than traditional capacitors as a result of their large-surface-area electrode materials and thin dielectrics [4]. Their unique properties, such as high power density, long life, and environmental protection characteristics, have attracted considerable attention. The electrode material is a key factor for improving supercapacitor performance [5,6]. The thickness of two-dimensional (2D) layered materials, which can reach the multi-atomic or even single-atomic level, is considerably smaller than that of materials with other dimensions. Therefore, these materials exhibit a large specific surface area, high mechanical toughness, and numerous surface active sites. These characteristics render 2D layered materials suitable for electrodes and energy-storage devices [7,8]. The main studied 2D layered electrode materials include graphene [9], metal oxide/hydroxide [10], and transition metal carbide/nitride (MXene) [11,12].

As a novel 2D layered material, MXene has received considerable research attention since its emergence [13,14]. Similar to the 2D structure and surface chemical composition of graphene, it exhibits good conductivity, hydrophilicity, excellent flexibility, and ion intercalation, presenting numerous application prospects for energy storage [15,16]. However, an in-depth study indicated that this material presents many limitations, such as easy agglomeration and stacking; the existence of surface groups substantially influences material properties [17]. The establishment of 3D porous structures is effective to prevent the stacking of MXene, completely use active sites available on the MXene surface, and improve material utilization. Li et al. prepared a 3D Ti_3_C_2_T_x_ aerogel with a unique channel and high specific capacitance of 1012.5 mF cm^−2^ for the mass loading of 15 mg at a scan rate of 2 mV s^−1^ in 1 M KOH electrolyte by employing an ethylenediamine-assisted self-assembly process [18]. Shi et al. successfully synthesized a Ti_3_C_2_T_x_ foam having a high capacitance of 122.7 F g^−1^ through the thermal treatment of Ti_3_C_2_T_x_ films with hydrazine monohydrate in 1 M KOH electrolyte at a scan rate of 5 mV s^−1^ [19]. Zhang et al. mixed Ti_3_C_2_T_x_ nanoparticles (etched using HF) and Ti_3_C_2_T_x_ nanosheets (etched with LiF and HCl) to acquire a flexible film with an open sandwich structure, showing a high specific capacitance of 372 F g^−1^ (1 A g^−1^), which was considerably higher than that of Ti_3_C_2_T_x_ films; after 5000 cycles, its cycle stability was as high as 95% [20]. However, these methods usually require a freeze–drying or filtration process, which limits their application in industrial production.

In recent years, Ti_3_C_2_T_x_ flakes with the mesoporous architecture, which show excellent electrode performance in sodium-ion batteries, have been prepared using the simple acid induced flocculation method of controlling the pH of Ti_3_C_2_T_x_ colloid solutions [21,22]. Dried Ti_3_C_2_T_x_ powders can be obtained using a simple centrifugation and drying process. However, this approach inevitably introduces undesirable –Cl ions onto the Ti_3_C_2_T_x_ surface. Studies have shown that alkalization [23,24] and annealing [25,26] can be performed to remove a part of surface groups and improve MXene electrochemical performance. In this study, we explored the possibility of preparing 3D porous Ti_3_C_2_T_x_ by using the alkaline-induced flocculation method. Furthermore, the alkalization flocculation effects of KOH, NaOH, and LiOH were compared in detail. Simultaneously, the effect of annealing on Ti_3_C_2_T_x_ properties after alkalization was studied. Moreover, the mechanism of alkalization flocculation and Ti_3_C_2_T_x_ surface modification was discussed.

## 2. Experimental and Procedure

### 2.1. Materials

Ti_3_AlC_2_ powders, used as raw materials, were synthesized through microwave sintering by using 3TiH_2_/1.2Al/2C mixtures [27]. Sintering temperature and time were 1300 °C and 30 min, respectively. Lithium fluoride (approximately 99.9 wt% pure) was purchased from Aladdin Reagent, Co., Ltd., Shanghai, China. All other chemicals, such as hydrochloric acid (HCl), potassium hydroxide (KOH), sodium hydroxide (NaOH), and lithium hydroxide (LiOH), obtained from Sinopharm Chemical Reagent Co., Ltd., Shanghai, China were of analytical grade.

### 2.2. Synthesis of Ti_3_C_2_T_x_ MXene

Ti_3_C_2_T_x_ MXene was obtained using the following methods. First, 2 g of LiF was dissolved in 40 mL of 6 M hydrochloric acid. The solution was stirred until the LiF powder was completely dissolved. Then, 2 g of the homemade Ti_3_AlC_2_ powder was slowly added to the LiF–HCl etching solution, and the resulting solution was maintained at 50 °C for 24 h under magnetic stirring. The solution was then centrifuged at 3000 rpm for 10 min to separate the powders from the supernatant and washed several times with deionized (DI) water until its pH was higher than six. Afterwards, the resulting Ti_3_C_2_T_x_ powder was added to 200 mL of DI water and ultrasonicated for 2 h in an ice bath and the Ar atmosphere. The resulting solution was centrifuged at 3000 rpm for 1 h. Finally, the dark green supernatant suspension, containing layered Ti_3_C_2_T_x_ flakes with a nominal concentration of approximately 1 mg mL^−1^, was obtained. We labelled these exfoliation Ti_3_C_2_T_x_ flakes as e-Ti_3_C_2_T_x_.

To form the 3D porous structure by adjusting the surface functional groups of Ti_3_C_2_T_x_, a common alkali, KOH, was used as a flocculent. The specific steps of alkalization were as follows: 0.7 g of 1 mol/L KOH was dissolved in 25 mL of e-Ti_3_C_2_T_x_ suspension. The mixed solution was allowed to stand for 30 min to complete alkalization. The supernatant was removed from the alkalization solution, and the remaining product was washed several times with DI water until pH became approximately seven. Then, the solution was centrifuged at 3000 rpm for 10 min to collect the sediments. The obtained sediment was dried at 60 °C for 12 h in a vacuum-drying oven. The alkalization sample treated with KOH is labelled as K-a-Ti_3_C_2_T_x_. For comparison, two other common alkalis (NaOH and LiOH) were used as flocculants, and their respective alkalization products were named Na-a-Ti_3_C_2_T_x_ and Li-a-Ti_3_C_2_T_x_. In addition, to improve the state surface functional groups of Ti_3_C_2_T_x_, after alkalization, M’-a-Ti_3_C_2_T_x_ (M’ represents K, Na, and Li) was annealed at 400 °C for 2 h in a vacuum tube furnace. The annealed samples are labelled as M’-A-Ti_3_C_2_T_x_ (M’ represents K, Na, and Li). The preparation process of 3D porous Ti_3_C_2_T_x_ is shown in Figure 1. To compare the performance between 3D porous M’-a-Ti_3_C_2_T_x_ obtained using alkalization and Ti_3_C_2_T_x_ acquired by employing the traditional pumping and filtration method, a flexible Ti_3_C_2_T_x_ paper (f-Ti_3_C_2_T_x_) was prepared, according to a method presented in the literature [28,29].

### 2.3. Structural Characterization

The phase composition of the sample was analyzed through X-ray diffraction (XRD) on a diffractometer (D8ADVANCE, Bruker, Karlsruhe, Germany) with Cu-Kα radiation at the step scan of 0.02° and 2θ of 5°–80°. Moreover, the surface chemical composition was studied through X-ray photoelectron spectroscopy (XPS, Axis Ultra DLD, Kyoto, Japan). The structure and morphology of the samples were analyzed using scanning electron microscopy (SEM, NovaNanoSEM450, FEI, Portland, OR, USA), energy dispersive spectroscopy (EDS, INCA Energy250 X-max 50, Oxford Instruments, FEI, Portland, OR, USA), and transmission electron microscopy (TEM, Tecnai™ G2 F30, FEI, Portland, OR, USA). The specific surface area of the samples was tested using the nitrogen adsorption and desorption method with the Beijing Precision Gaobo JW-BK132F automatic specific surface analyzer. The experimental pretreatment temperature and time were 300 °C and 2 h, respectively.

### 2.4. Electrochemical Measurements

The electrochemical properties were evaluated at room temperature in a traditional three-electrode system and a 1 M H_2_SO_4_ electrolyte. Anodes were prepared by mixing the as-prepared dry Ti_3_C_2_T_x_ powders, acetylene black, and polytetrafluoroethylene (PTFE) in a ratio of 8:1:1 in a mortar and pestle. Anhydrous ethanol was used as the solvent to obtain a uniform slurry. The resulting slurry was evenly coated on a stainless steel net and dried at 60 °C for 12 h in the vacuum oven. The as-prepared anodes, a platinum foil electrode, and a saturated Hg/HgSO_4_ electrode served as the working, counter, and reference electrodes, respectively. Cyclic voltammetry (CV), galvanostatic charge–discharge (GCD) studies, and electrochemical impedance spectroscopy (EIS) were conducted on an electrochemical workstation (CHI660E, Shanghai Chenhua Instrument Co., Ltd., Shanghai, China). Electrochemical impedance spectroscopy (EIS) was performed at the open-circuit potential with an amplitude of 5 mV in a frequency range of kHz to 0.01 Hz.

## 3. Results and Discussion

The appearance of the samples is shown in Figure 2. The Ti_3_AlC_2_ powder is greyish black (Figure 2a). After etching and stripping, the suspension solution turns dark green and shows high light transmittance (Figure 2b), which proves that the suspensions solution of e-Ti_3_C_2_T_x_ exhibits uniform dispersion. The f-Ti_3_C_2_T_x_ paper obtained by pumping and filtering presents a high flexibility (Figure 2c), which is consistent with previous reports [28,29]. After the addition of an appropriate amount of alkalis, such as KOH, NaOH, and LiOH, to the e-Ti_3_C_2_T_x_ suspension solution, the e-Ti_3_C_2_T_x_ nanoflakes can undergo flocculation and gradually aggregate at the bottom of the container (Figure 2d). However, the flocculation degree of M-a-Ti_3_C_2_T_x_ reveals that the flocculating rate was alkalized; the first is NaOH, then is KOH, and LiOH. This finding is consistent with the trend of the decreasing radius of alkali metal ions. Therefore, we speculate that alkali metal ions play a crucial role in flocculation.

Figure 3a–h shows the XRD patterns of Ti_3_AlC_2_, f-Ti_3_C_2_T_x_, K-a-Ti_3_C_2_T_x_, Na-a-Ti_3_C_2_T_x_, Li-a-Ti_3_C_2_T_x_, K-A-Ti_3_C_2_T_x_, Na-A-Ti_3_C_2_T_x_, and Li-A-Ti_3_C_2_T_x_, respectively. The right side of Figure 3 presents the enlarged XRD pattern with diffraction angle of 5°–10°. The characteristic peaks of Ti_3_AlC_2_ completely disappear after etching (Figure 3), which indicates that Ti_3_AlC_2_ is completely converted into Ti_3_C_2_Tx. For f-Ti_3_C_2_T_x_, only one (002) characteristic peak located at 6.21° is observed, and calculated d_c_/2, considered to be half of the c lattice parameter, is 14.2 Å. However, except f-Ti_3_C_2_T_x_, all other treated samples exhibit two characteristic diffraction peaks at 34.7° (101) and 61.3° (110). This finding indicates that the f-Ti_3_C_2_T_x_ film flakes are highly ordered parallel to the substrate. In addition, the (002) peaks of the samples obtained through alkalization with KOH, NaOH, and LiOH appear at 6.66°, 6.36°, and 5.79°, respectively. Their corresponding dc/2 values are 13.2, 13.8, and 15.2 Å, respectively, which are larger than those of Ti_3_AlC_2_. K-a-Ti_3_C_2_T_x_ has smaller d_c_/2 than Na-a-Ti_3_C_2_T_x_ and Li-a-Ti_3_C_2_T_x_. This trend is inconsistent with the increase in ion radii reported in the literature [30]; however, it is positively correlated with the trend of hydrated ion radii [31]. Therefore, we speculated that the water molecules can co-embed with alkali metal ions during alkalization. The (002) peaks of K-A-Ti_3_C_2_T_x_, Na-A-Ti_3_C_2_T_x_, and Li-A-Ti_3_C_2_T_x_ shifted towards right. The calculated d_c_/2 values of K-A-Ti_3_C_2_T_x_, Na-A-Ti_3_C_2_T_x_, and Li-A-Ti_3_C_2_T_x_ are 12.2, 13.1, and 13.5 Å, respectively. The layer spacing of the samples obviously decreased after annealing. To some extent, this phenomenon may occur as annealing can lead to the purification of sample surface ionic groups and crystal water. Simultaneously, due to the oxidation of Ti_3_C_2_T_x_, a small amount of the TiO_2_ phase is inevitably found in the annealed samples [32].

To confirm the aforementioned speculation, the XPS spectra of f-Ti_3_C_2_T_x_ and K-a-Ti_3_C_2_T_x_ were analyzed. The XPS results are shown in Figure 4. The complete XPS spectrum (Figure 4a) suggests the presence of C, Ti, O, F, and a small amount of Cl elements in f-Ti_3_C_2_T_x_. In K-a-Ti_3_C_2_T_x_ obtained through KOH alkalization, the strength of F and Cl elements is weaker than that in f-Ti_3_C_2_T_x_. An extra element (K) is observed in K-a-Ti_3_C_2_T_x_ after KOH alkalization. Therefore, the content of –F and –Cl ions on the Ti_3_C_2_T_x_ surface can be effectively reduced through alkalization. In addition, in K-A-Ti_3_C_2_T_x_ (Figure 4a), the Cl element almost disappears after annealing, and the intensity of element F considerably decreases. The annealing process can be further used to reduce the content of –F and –Cl ions on the Ti_3_C_2_T_x_ surface.

The high-resolution XPS spectra of C 1s (Figure 4b) presents the peaks for surface Ti–C (281.8 eV), Ti–C–O (282.4 eV), C–C (284.8 eV), C–O–C (286.4 eV), and O–C=O (288.3 eV). Three peaks of the O 1s XPS spectra are attributed to surface Ti–O (529.8 eV), Ti–O–H (531.6 eV), and C–O (532.4 eV) (Figure 4c). Six peaks of Ti–C 2p_3/2_ (454.7 eV), Ti(II) 2p_2/3_ (555.8 eV), Ti–O 2p_2/3_ (558.3 eV), Ti–C 2p_1/2_ (460.8 eV), Ti(II) 2p_1/2_ (461.8 eV), and Ti-O 2p_1/2_ (464.0 eV) appear in the Ti 2p XPS spectra (Figure 4d). The results of C 1s, O 1s, and Ti 2p indicate that the peak intensities of C–O–C, O–C=O, and Ti–O 2p_2/3_ enhance after alkalization. The vacancy created by the reduction in –F and –Cl groups during alkalization is occupied by the –OH or =O group. After annealing, the peak strength of Ti–O and Ti–O 2p_2/3_ increases. The small amount of exposed Ti, obtained from the removal of surface groups such as –F and –Cl during alkalization and annealing, is oxidized, which is consistent with the XRD results presented in Figure 3f. The peak strength of Ti–C decreased, indicating that the oxidation of Ti leads to the fracture of a part of the Ti–C bond. The Ti–O–H bond disappears after annealing and is replaced with the =O bond. This phenomenon has positive practical significance for improving the performance of Ti_3_C_2_T_x_ [33,34].

The morphology and microstructure of Ti_3_AlC_2_, e-Ti_3_C_2_T_x_, f-Ti_3_C_2_T_x_, K-a-Ti_3_C_2_T_x_, Na-a-Ti_3_C_2_T_x_, Li-a-Ti_3_C_2_T_x_, K-A-Ti_3_C_2_T_x_, Na-A-Ti_3_C_2_T_x_, and Li-A-Ti_3_C_2_T_x_ were observed through SEM (Figure 5). Ti_3_AlC_2_ exhibits the typical lamellar structure of MAX phase ceramic materials (Figure 5a). The warping edges of e-Ti_3_C_2_T_x_ nanosheets can be observed (Figure 5b). These results show that the multilayer Ti_3_C_2_T_x_ becomes completely stripped after dispersion stripping, and the obtained e-Ti_3_C_2_T_x_ nanosheets exhibit few layers or a monolayer with flexibility. Compared with that of e-Ti_3_C_2_T_x_, the cross-sectional view of f-Ti_3_C_2_T_x_ paper shows a well-arranged, layered, and extremely coherent structure (Figure 5c). This finding can further verify the XRD results shown in Figure 3b. However, K-a-Ti_3_C_2_T_x_ with the 3D porous structure and fold shape was prepared through alkaline-induced flocculation by using KOH (Figure 5d). The 3D porous structure comprises crumpled, open, interpenetrating, Ti_3_C_2_T_x_ nanosheets, which form many irregular large pores with diameters of 100−500 nm. Compared with KaOH utilisation, when NaOH or LiOH (Figure 5d,e) was used for alkalinisation, no obvious differences were observed in the final 3D porous structure. After annealing, the 3D porous structures of the samples were not damaged (Figure 5g–i). However, a small amount of particles attached to the M’-A-Ti_3_C_2_T_x_ flakes, M’ represents K, Na, and Li. The XRD results (Figure 3f–h) reveal that these small particles are TiO_2_.

The TEM images of f-Ti_3_C_2_T_x_, K-a-Ti_3_C_2_T_x_, and K-A-Ti_3_C_2_T_x_ are shown in Figure 6a–c. The thin e-Ti_3_C_2_T_x_ flake has a graphene-like structure (Figure 6a). After alkalization, the Ti_3_C_2_T_x_ flakes curls and folds into the 3D porous structure (Figure 6b). This finding is consistent with the SEM results shown in Figure 5d. The high-resolution TEM (HRTEM) results reveal that the lattice spacing of the sample after alkalization decreases from 1.41 to 1.31 nm. After annealing, the lattice spacing further decreases to approximately 1.22 nm, which is consistent with the XRD results.

The N_2_ adsorption–desorption isotherms are shown in Figure 7. The curves of several samples show obvious H3 hysteresis loops for the relative pressure of 0.5–1, which indicates that the layered material aggregates have mesopores and macropores. The specific surface areas of f-Ti_3_C_2_T_x_, Li-a-Ti_3_C_2_T_x_, Na-a-Ti_3_C_2_T_x_, K-a-Ti_3_C_2_T_x_ and K-A-Ti_3_C_2_T_x_ are 2.21, 10.51, 19.67, 30.73 and 33.06 m^2^ g^−1^, respectively. The specific surface areas of the samples before the alkalization–flocculation treatment are small as a result of the stacking and agglomeration of nanosheets. The specific surface areas considerably increased after the alkalization–flocculation treatment, whereas after annealing treatment, the surface area only slightly increased. The increase in the specific surface area considerably affects electrochemical performance.

The electrochemical performance of the samples was tested in the three-electrode system. Figure 8a shows the CV curves of all the samples at a scan rate of 10 mV s^−1^. After alkalization, the CV curve area of the sample increases. The area further increases after annealing. The specific capacitance of the material is proportional to the area under the CV curve. The capacitance of the 3D porous material obtained through alkalization and annealing increases considerably. The CV curves of f-Ti_3_C_2_T_x_ and a-Ti_3_C_2_T_x_ are nearly rectangular; however, a certain degree of distortion exists in the shape, which indicates that the double-layer charge and discharge are accompanied by Faraday redox processes in the sulfuric acid electrolyte. The bonding/debonding behavior can be observed between hydrated hydrogen and terminal oxygen for the Ti_3_C_2_T_x_ electrode. This behavior changes the valence state of Ti and generates a pseudocapacitance [35]. After annealing, the distortion degree of the CV curve of the sample aggravates, and a pair of redox peaks can be observed, which is attributed to the appearance of TiO_2_.

The GCD discharge time is usually used to represent the electrode capacity to store charges at the same current density. Figure 8b shows the GCD curve of all the samples at a current density of 1 A g^−1^. According to the GCD curves (Figure 8b), K-A-Ti_3_C_2_T_x_ has the longest discharge time, indicating that it has the highest specific capacity. The EIS test was performed on the electrode material. The measured Nyquist curve is shown in Figure 8c. The Nyquist curve mainly includes two parts: the high- and low-frequency areas. The enlarged image of the high-frequency area (Figure 8c) shows that Ti_3_C_2_T_x_ has a smaller semicircle; thus, the charge transfer resistance is small, which contributes to high magnification performance. The curve in the low-frequency region appears as a straight line. Its slope corresponds to the capacitance characteristics of the material, which is related to ion diffusion/transmission and Warburg impedance. The higher is the slope value of the curve, the lower is the diffusion resistance [36]. The straight lines of all the samples are almost perpendicular to the X axis, indicating good capacitance characteristics. The slope of K-A-Ti_3_C_2_T_x_ is the highest, indicating that it has better ion mobility and lower diffusion resistance than the other samples.

Figure 8d,e shows the CV and GCD curves of K-A-Ti_3_C_2_T_x_ at different scan rates and current densities, respectively. When the scan rate increases, the shape of the CV curve shows a certain degree of distortion, which is related to the low diffusion of electrolyte ions at high scan rates [37]. The charging and discharging times of the GVD curve are equal, and the shape of the curve is close to an isosceles triangle, indicating that the material exhibits high Coulomb efficiency and good capacitance characteristics. The cycle stability test was performed on K-A-Ti_3_C_2_T_x_. After 5000 charge–discharge cycles at the current density of 5 A g^−1^, the specific capacity remained 89% of the initial value. Figure 8f presents the GCD curve for the 1st and 5000th cycles. The shape of the GCD curve for the sample of K-A-Ti_3_C_2_T_x_ remains excellent, indicating that the material exhibits good reversibility and cycle stability.

Figure 9 shows the specific capacitance of Ti_3_C_2_T_x_ electrodes at the scan rates of 1–20 A g^−1^. At the current density of 1 A g^−1^, the mass specific capacities of f-Ti_3_C_2_T_x_, K-a-Ti_3_C_2_T_x_, Na-a-Ti_3_C_2_T_x_, Li-a-Ti_3_C_2_T_x_, K-A-Ti_3_C_2_T_x_, Na-A-Ti_3_C_2_T_x_, and Li-A- Ti_3_C_2_T_x_ are 228.2, 300.2, 285.7, 250.8, 400.7, 343.3, and 326.7, respectively. The electrochemical performance of K-A-Ti_3_C_2_T_x_ improved substantially. The subsequent annealing treatment further reduced the content of unfavorable surface functional groups, and electrochemical performance was improved. This result is higher than previous report on Ti_3_C_2_T_x_-based materials, the detailed data are listed in Table 1. The specific capacities of the Ti_3_C_2_T_x_ samples obtained through alkaline flocculation and annealing at different current densities are excellent. With its excellent 3D porous structure, perfect surface cleanliness, and large layer spacing, K-A-Ti_3_C_2_T_x_ shows outstanding electrochemical performance.

## 4. Conclusions

In summary, this study confirmed that the addition of alkali to Ti_3_C_2_T_x_ colloids can make nanosheets to flocculate and deposit. Alkalization and flocculation were conducted simultaneously, and the 3D porous structure was constructed through simple and effective means. The pore diameter of macropores can reach several hundred nanometres, which considerably increases the specific surface area of the material. Alkalization and annealing effectively reduced the adverse surface groups of the material and exposed more active sites, which is beneficial for improving the electrochemical performance of the material. The sample treated with KOH exhibits a higher specific surface area (33.06 m^2^ g^−1^) and better electrochemical performance than that treated with NaOH and LiOH. In the three-electrode system with 1 M H_2_SO_4_ as the electrolyte, K-A-Ti_3_C_2_T_x_ shows a high specific capacity of approximately 400.7 F g^−1^ at the current density of 1 A g^−1^ and attains 89% cycle stability after 5000 charge–discharge cycles. A simple and effective strategy can promote the practical application of Ti_3_C_2_T_x_ supercapacitor electrodes.

## Figures and Tables

**Figure 1 materials-15-00925-f001:**
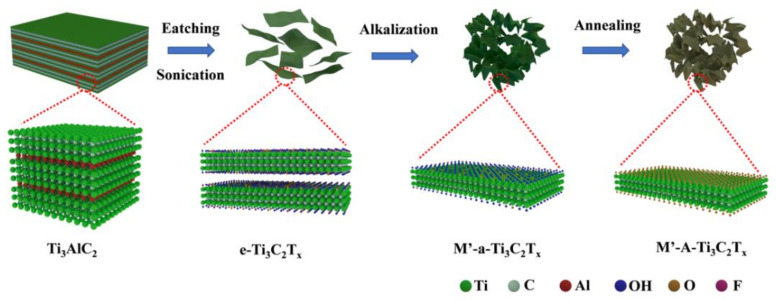
Preparation of 3D porous Ti_3_C_2_T_x_.

**Figure 2 materials-15-00925-f002:**
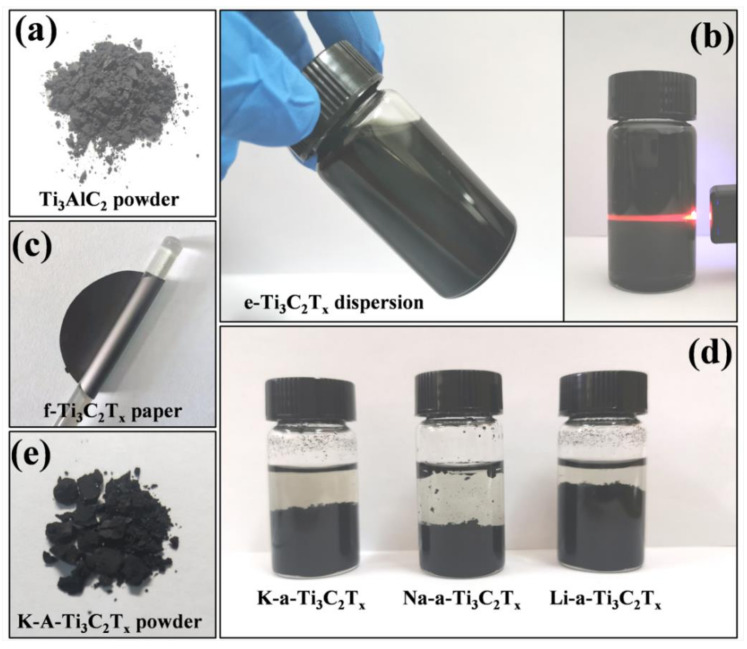
Appearance of the samples: (**a**) Ti_3_AlC_2_ powder, (**b**) e-Ti_3_C_2_T_x_, (**c**) f- Ti_3_C_2_T_x_ paper, (**d**) K-a-Ti_3_C_2_T_x_, Na-a-Ti_3_C_2_T_x_ and Li-a-Ti_3_C_2_T_x_, and (**e**) K-A-Ti_3_C_2_T_x_ powder.

**Figure 3 materials-15-00925-f003:**
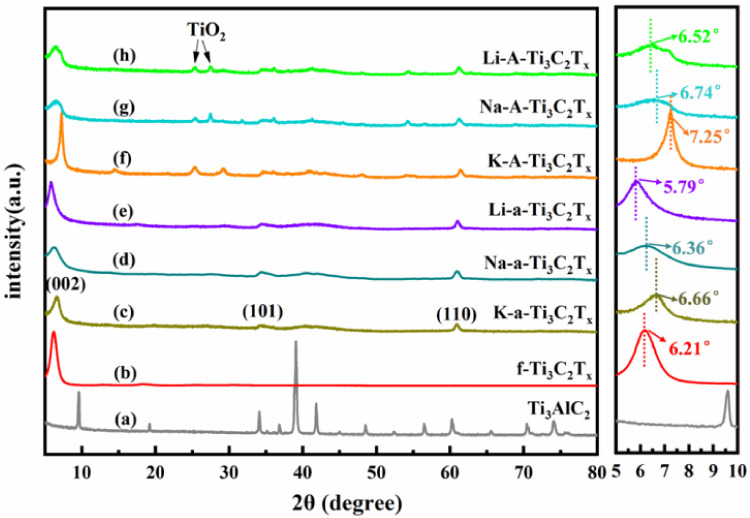
XRD patterns of (**a**) Ti_3_AlC_2_, (**b**) f-Ti_3_C_2_T_x_, (**c**) K-a-Ti_3_C_2_T_x_, (**d**) Na-a-Ti_3_C_2_T_x_, (**e**) Li-a-Ti_3_C_2_Tx, (**f**) K-A-Ti_3_C_2_T_x_, (**g**) Na-A-Ti_3_C_2_T_x_, and (**h**) Li-A-Ti_3_C_2_T_x_. The right side of the figure presents the enlarged XRD pattern with diffraction angle of 5°–10°.

**Figure 4 materials-15-00925-f004:**
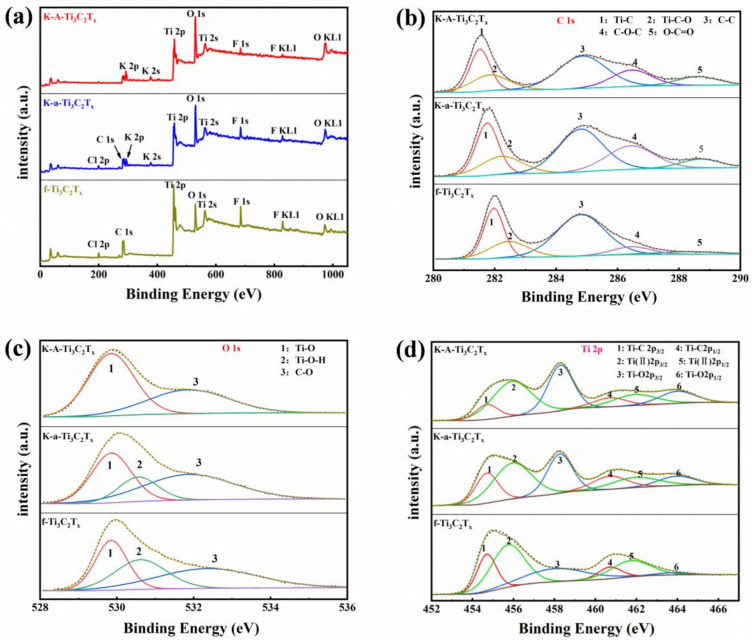
XPS results of f-Ti_3_C_2_T_x_, K-a-Ti_3_C_2_T_x_, and K-A-Ti_3_C_2_T_x_: (**a**) full spectrum, (**b**) C 1s spectra, (**c**) O 1s spectra, and (**d**) Ti 2p spectra.

**Figure 5 materials-15-00925-f005:**
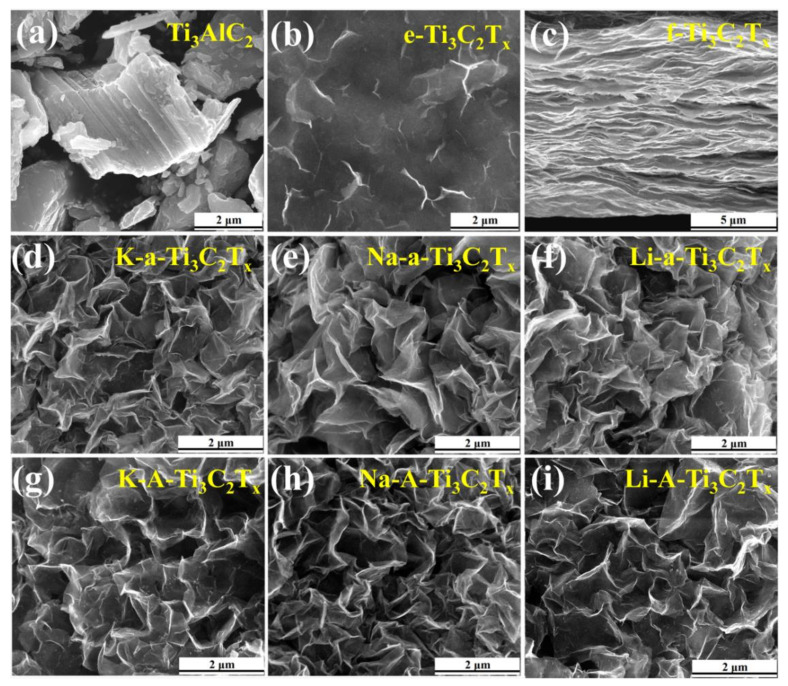
SEM images of: (**a**) Ti_3_AlC_2_, (**b**) e-Ti_3_C_2_T_x_, (**c**) f-Ti_3_C_2_T_x_, (**d**) K-a-Ti_3_C_2_T_x_, (**e**) Na-a-Ti_3_C_2_T_x_, (**f**) Li-a-Ti_3_C_2_T_x_, (**g**) K-A-Ti_3_C_2_T_x_, (**h**) Na-A-Ti_3_C_2_T_x_, and (**i**) Li-A-Ti_3_C_2_T_x_.

**Figure 6 materials-15-00925-f006:**
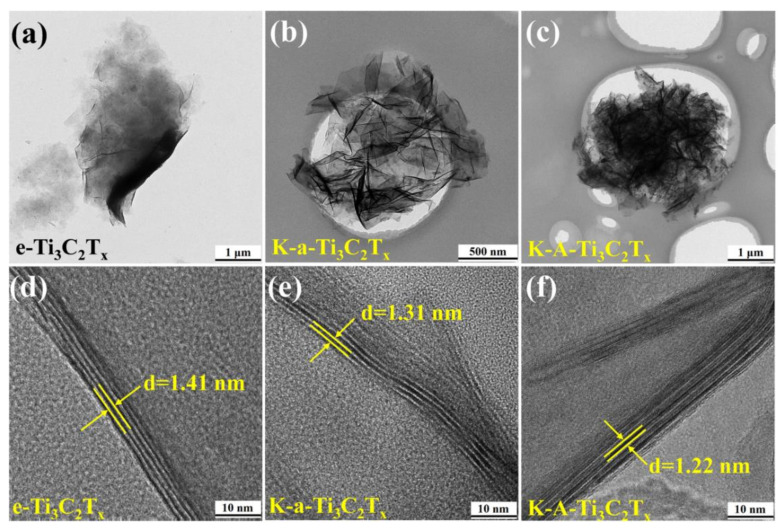
TEM images of: (**a**) e-Ti_3_C_2_T_x_, (**b**) K-a-Ti_3_C_2_T_x_, (**c**) K-A-Ti_3_C_2_T_x_ and HRTEM images of: (**d**) e-Ti_3_C_2_T_x_, (**e**) K-a-Ti_3_C_2_T_x_, and (**f**) K-A-Ti_3_C_2_T_x_.

**Figure 7 materials-15-00925-f007:**
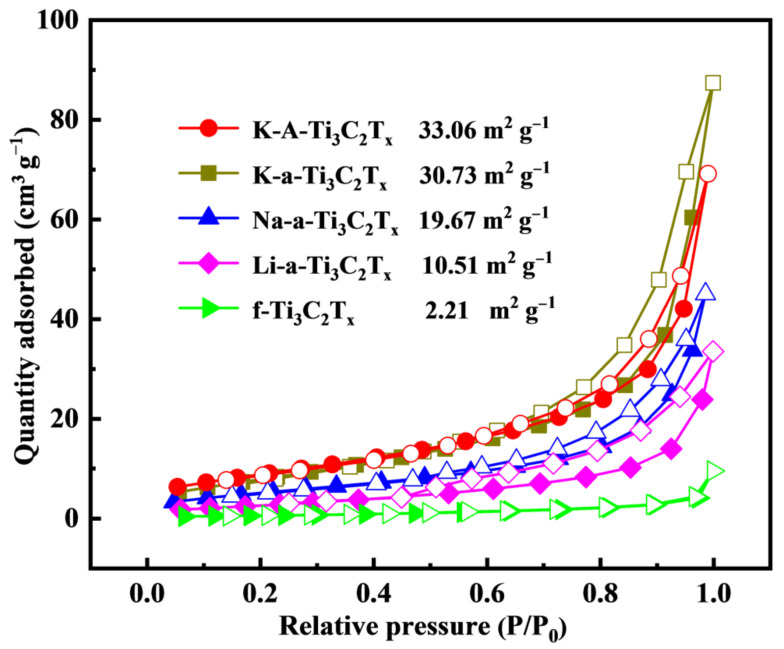
N_2_ adsorption–desorption isotherms of f-Ti_3_C_2_T_x_, K-a-Ti_3_C_2_T_x_, Na-a-Ti_3_C_2_T_x_, Li-a-Ti_3_C_2_T_x_, and K-A-Ti_3_C_2_T_x_.

**Figure 8 materials-15-00925-f008:**
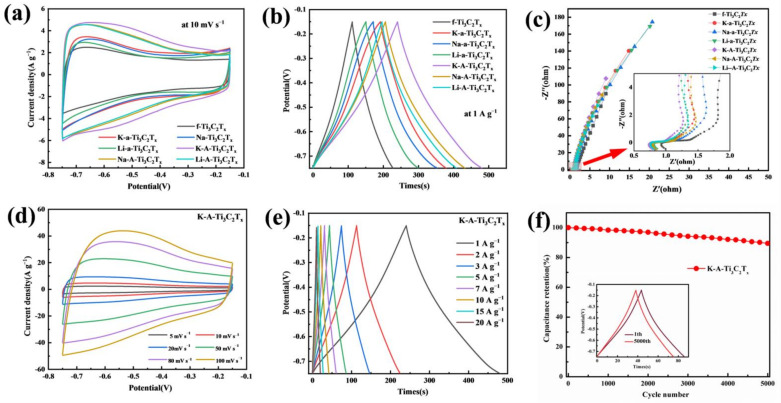
(**a**) CV curves of the Ti_3_C_2_T_x_ electrodes at the scan rate of 10 mV s^−1^, (**b**) GCD profiles of the Ti_3_C_2_T_x_ electrodes at the scan rate of 1 A g^−1^, (**c**) Electrochemical impedance spectroscopy data of the Ti_3_C_2_T_x_ electrodes, (**d**) CV curves of the K-A-Ti_3_C_2_T_x_ electrode at the scan rates of 10–100 mV s^−1^, (**e**) GCD profiles of the K-A-Ti_3_C_2_T_x_ electrode at the scan rates of 1–20 A g^−1^, and (**f**) Cyclic stability of the K-A-Ti_3_C_2_T_x_ electrode at a current density of 5 A g^−1^.

**Figure 9 materials-15-00925-f009:**
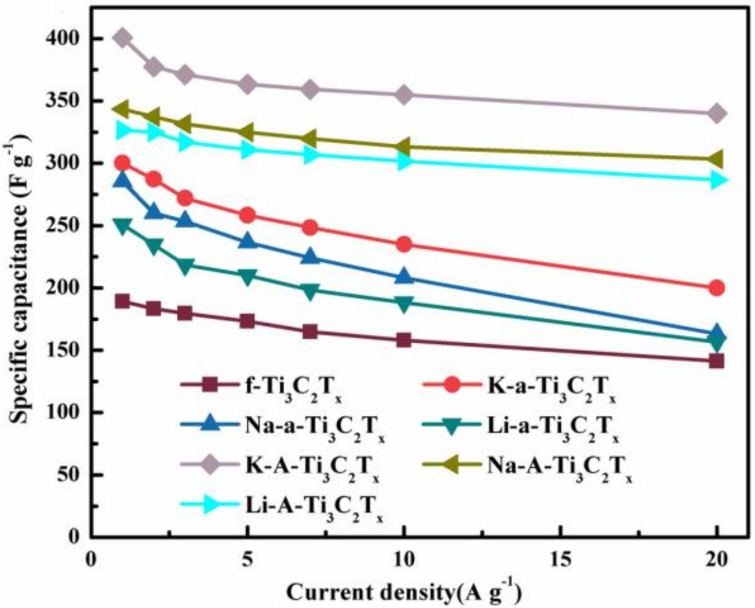
Specific capacitance of the Ti_3_C_2_T_x_ electrode at the scan rates of 1–20 A g^−1^.

**Table 1 materials-15-00925-t001:** Comparison of capacitance for the Ti_3_C_2_T_x_-based electrodes materials.

**Material**	**Electrolyte**	**Scan Rate**	**Capacitance**	**Ref.**
Ti_3_C_2_T_x_-foam	1 M KOH	5 mV S^−1^	122.7 F g^−1^	[19]
flexible Ti_3_C_2_T_x_	1 M H_2_SO_4_	1 A g^−1^	372 F g^−1^	[20]
Ti_3_C_2_T_x_	1 M H_2_SO_4_	2 mV S^−1^	245 F g^−1^	[38]
porous Ti_3_C_2_T_x_ film	3 M H_2_SO_4_	10 V S^−1^	210 F g^−1^	[39]
N-doped Ti_3_C_2_T_x_	1 M H_2_SO_4_	1 mV S^−1^	192 F g^−1^	[11]
Graphene/Ti_3_C_2_T_x_	1 M H_2_SO_4_	10 mV S^−1^	327.5 F g^−1^	[40]
K-a-Ti_3_C_2_T_x_	1 M H_2_SO_4_	1 A g^−1^	300.2 F g^−1^	This work
K-A-Ti_3_C_2_T_x_	1 M H_2_SO_4_	1 A g^−1^	400.7 F g^−1^	This work

## Data Availability

Data are contained within the article and can be requested from the corresponding author.

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
