# Peer review of "3D Porous MXene (Ti3C2Tx) Prepared by Alkaline-Induced Flocculation for Supercapacitor Electrodes"

_materials, 2022, doi:10.3390/ma15030925_

Round 1

Reviewer 1 Report

The work on 3D porous MXene for supercapactior electrodes looks interesting. The standard of English is very poor. Some sentences are difficult to understand. So, I suggest doing extensive English editing. I have highlighted  few sentences in yellow. 

In figure 6, the legend and the figure caption nomenclature is not the same. Otherwise, seems good to be published in Materials.

Reviewer 2 Report

The authors must proof read their manuscript carefully, preferably by a native English speaker - a major task that was evidently lacking - as the manuscript is full of grammatical errors and incomprehensible sentence structures.

While there are some novelties in this research where the authors used alkaline treated MXene powders to produce porous structures,  the   improvement over earlier studies is not hugely significant.   Thus I consider it a borderline case for Materials.

Reviewer 3 Report

Adequate/optimum design and manufacturing of energy storage devices is an important scientific/technological task related directly to the commercial applications of the devices. Indeed, the increasing global demand of sustainable energy supply requires more efficient energy storage with high energy and power densities. Supercapacitors are making their way in different commercial applications. Analysis and optimization of their materials and design parameters are of crucial importance to improve their performance. Authors present an improved method to obtain 3D porous structures from MXene (Ti3C2Tx). In this method, authors have performed alkaline induced flocculation, using KOH, NaOH and LiOH, and annealing of the samples. As a result, they have obtained a good specific capacitance and a good retention rate of capacitance.

First, I have two general comments:

1.- Introduction is somewhat confusing for me. It seems that 2D materials are presented as good materials for supercapacitors. As far as I know, these materials can not be directly used as active materials. However, they can be good precursors to obtain 3D structured/porous materials.

Moreover, “The thickness of the two-dimensional layered material is much smaller than the other two dimensions”. The thickness of the active material of electrodes is smaller than the other two dimensions of the electrode. Nevertheless, I think authors are speaking about material and not electrodes. Moreover, in supercapacitors, mainly porous/disordered carbons are used to increase the specific capacitance, for example, activated carbons. These materials have a 3D disordered structure. Therefore, these explanations are confusing for me.

2.- Some figures show too much information (for example, Fig. 7 and Fig. 8 c)). It is difficult to analyse the behaviour of each material.

Second, I have also some little comments about details of the manuscript that are not clear for me…

(Page 1, abstract) “the sample alkaline treatment by KOH (K-A-Ti3C2Tx) has an excellent capacity about 401 F g-1”. Capacity or capacitance? They are very different things. This same mistake appear in different places of the manuscript.

(Page 1, abstract) “The work may provide an effective way to preparation of high performance supercapacitors electrode”. Electrode or materials?

(Page 1) “The potential depletion of fossil energy”. At this moment, I don’t think that fossil fuels are going to finish in a short period of time. It seems that we have passed the peak oil. However, there is plenty of fossil fuels for at least two generations. In my opinion, the problem is another one.

(Page 1) “Its unique properties such as high power”-DENSITY?

(Page 2) Ref. 11 is devoted to graphene-based supercapacitors. I don’t think it is adequate as an example for transition metal carbide/nitride materials.

(Page 2) The capacitance obtained by Li et al. (Ref. 18) has to be given for comparison, as the other ones.

(Page 8) Which are the specific surface areas of Na-A-Ti3C2Tx, Li-A-Ti3C2Tx? How do you explain the high differences between the specific areas of K-a-Ti3C2Tx, Na-a-Ti3C2Tx and Li-a-Ti3C2Tx?

(Page 11) The “amazing electrochemical performance” of K-A-Ti3C2Tx should make it adequate for commercial applications. However, in the manuscript, it should be interesting to give some values of commercial electrode materials for comparison. Which are the usual capacitance values of commercial materials of supercapacitors? Moreover, which could be the possible problems of the commercial use of K-A-Ti3C2Tx? Could it be manufactured in an industrial way without problems?

Finally, there are some points related to the correct writing of the manuscript that should be improved:

(Page 1) “… with good conductivity and pseudo-capacitance properties IS shows great application potential”.

(Page 1) “… are relatively lowly”.

(Page 1) “which are higher capacitance than traditional capacitors due to use high surface area electrode materials and thin dielectrics to achieve”. It seems that something is missing.

(Page 3) “The schematic diagram of preparation process 3D porous Ti3C2Tx is showN in Figure. 1”.

(Page 4) “The appearance of the samples is shown AS Figure 2”.

(Page 8) “the morphology of thin e-Ti3C2Tx flake has a graphene-like.”

(Page 11) “In summary, this work PROVES that adding alkali to Ti3C2Tx colloid CAN MAKE nanosheets flocculate and deposit.”

Round 2

Reviewer 2 Report

The manuscript is now in much better shape for publishing.